# Magnetic Resonance Imaging of Macrophage Response to Radiation Therapy

**DOI:** 10.3390/cancers15245874

**Published:** 2023-12-17

**Authors:** Harrison Yang, Brock Howerton, Logan Brown, Tadahide Izumi, Dennis Cheek, J. Anthony Brandon, Francesc Marti, Roberto Gedaly, Reuben Adatorwovor, Fanny Chapelin

**Affiliations:** 1F. Joseph Halcomb III, M.D. Department of Biomedical Engineering, University of Kentucky, Lexington, KY 40506, USA; hsya222@uky.edu (H.Y.); labrown9822@gmail.com (L.B.); 2Shu Chien-Gene Lay Department of Bioengineering, University of California San Diego, La Jolla, CA 92093, USA; b1howerton@ucsd.edu; 3Lucille Parker Markey Cancer Center, University of Kentucky, Lexington, KY 40536, USA; t.izumi@uky.edu (T.I.); fmart3@email.uky.edu (F.M.); rgeda2@uky.edu (R.G.); 4Department of Toxicology and Cancer Biology, University of Kentucky, Lexington, KY 40536, USA; 5Department of Radiation Medicine, University of Kentucky, Lexington, KY 40536, USA; dennis.cheek@uky.edu; 6Sanders Brown Center on Aging, University of Kentucky, Lexington, KY 40508, USA; j.anthony.brandon@uky.edu; 7Department of Surgery, Transplant Division, University of Kentucky, Lexington, KY 40506, USA; 8Department of Biostatistics, University of Kentucky, Lexington, KY 40536, USA; radatorwovor@uky.edu; 9Department of Radiology, University of California San Diego, La Jolla, CA 92093, USA

**Keywords:** magnetic resonance imaging, cancer, radiation therapy, fluorine, macrophage, tumor microenvironment

## Abstract

**Simple Summary:**

The bodily response to cancer treatment is difficult to track in real time. This study aimed to demonstrate the possible application of magnetic resonance imaging (MRI) as a non-invasive and non-radiative modality to evaluate cancer response to therapy. We injected a fluorinated contrast agent in mouse models of breast and colon cancer to tag and track macrophages in real time. Longitudinal tracking of macrophages in tumors following radiation therapy via MRI provided a means of analyzing the tumor microenvironment without the need for a biopsy. It was shown that tumor-associated macrophage recruitment can be quantified through magnetic resonance imaging. Moreover, the findings suggest that macrophage response to radiation therapy is dependent on several factors including tumor origin. Since tumor recurrence following therapy is a major barrier following treatment, this imaging technique will be extremely beneficial in the future analysis of inflammation’s role in that process.

**Abstract:**

Background: Magnetic resonance imaging (MRI) is a non-invasive imaging modality which, in conjunction with biopsies, provide a qualitative assessment of tumor response to treatment. Intravenous injection of contrast agents such as fluorine (^19^F) nanoemulsions labels systemic macrophages, which can, then, be tracked in real time with MRI. This method can provide quantifiable insights into the behavior of tumor-associated macrophages (TAMs) in the tumor microenvironment and macrophage recruitment during therapy. Methods: Female mice received mammary fat pad injections of murine breast or colon cancer cell lines. The mice then received an intravenous ^19^F nanoemulsion injection, followed by a baseline ^19^F MRI. For each cancer model, half of the mice then received 8 Gy of localized radiation therapy (RT), while others remained untreated. The mice were monitored for two weeks for tumor growth and ^9^F signal using MRI. Results: Across both cohorts, the RT-treated groups presented significant tumor growth reduction or arrest, contrary to the untreated groups. Similarly, the fluorine signal in treated groups increased significantly as early as four days post therapy. The fluorine signal change correlated to tumor volumes irrespective of time. Conclusion: These results demonstrate the potential of ^19^F MRI to non-invasively track macrophages during radiation therapy and its prognostic value with regard to tumor growth.

## 1. Introduction

Inflammation is well-known for its critical role in many bodily functions such as the defense against pathogens or promoting tissue repair and regeneration [1]. Within the past few decades, cancer research has become more holistic, realizing inflammatory processes play a critical role in both tumor development and eradication [2,3]. These contrasting inflammatory processes have been defined as anti-tumorigenic inflammation, that suppresses tumor growth, and pro-tumorigenic inflammation, that benefits growth and exerts tumor-promoting signals onto both epithelial and cancer cells [2]. However, there is uncertainty determining a balance between these complex interactions.

Surgery, chemotherapy, and radiotherapy (RT) are all mainstay modalities for treating cancer. The body responds to these therapies through the immune responses, and distinguishing between antitumor and tumor-promoting immune responses can be difficult [3]. Radiotherapy eradicates cancer cells through necrosis, a pro-inflammatory form of cell death [3,4]. The inflammation that results from necrosis possesses the ability to promote rapid tumor growth or enhance the cross-presentation of tumor antigens, which subsequently induces an antitumor response [3].

Radiation therapy (RT) is used in over half of all cancer patients [5]. RT is localized and more cost-effective than other cancer treatments, but often fails to fully eradicate cancer. Both breast and colorectal cancers present a high incidence worldwide [6,7,8]. In 2020, breast cancer surpassed lung cancer to become the most commonly diagnosed cancer in the world irrespective of gender [7,8]. Meanwhile, colon cancer frequently ranks within the top five in several studies [6,7,8]. In early stage (stage I–II) breast cancer, RT is primarily used as an adjuvant therapy to reduce mortality following breast-conserving surgery [9,10,11,12,13]. A 2017 study determined that 78.4% of breast cancer patients undergo RT [14]. Another 2019 study found that 39.8% of patients possessing colorectal cancer are also treated with RT [15]. Although these modalities for eradicating cancer often show initial success, tumor recurrence still acts as a major barrier against complete remission. By gaining a deeper insight into the tumor microenvironment (TME) through a non-invasive biomarker, we may attain a more definitive understanding on tumor relapse following radiation therapy. A focused examination of the interactions, timing, and underlying factors responsible for macrophage polarization could yield valuable information.

Tumor endothelial cells are the most sensitive to radiation, leaving the tumor vasculature impaired and stimulating bone-marrow-derived cells and pro-inflammatory macrophages to infiltrate the remaining tumor tissue and reorganize the local vasculature [16]. These cells can later differentiate to the M2 macrophage subtype, promoting early recurrence [16,17,18] as established by histopathology analyses. The standard practice to determine TAM or macrophage burden involves the immunostaining of biopsied tumor specimens [19,20,21]. This technique is fraught with numerous limitations, including the invasive nature of the test, the heterogeneous TAM distribution in primary and metastatic lesions, and the lack of a clear cut-off point for positive staining [22,23,24].

An imaging biomarker capable of non-invasively depicting and quantifying tumor-associated macrophages and de novo macrophages infiltrating the tumor in vivo may be essential for an accurate prognosis assessment following radiation therapy. Macrophages have been labeled with various contrast agents and tracked via magnetic resonance imaging (MRI) in many disease models [25,26,27,28,29,30,31,32,33,34,35,36,37,38], yet sparsely for radiation therapy or clinical applications [39,40,41]. Fluorine-based formulations have continually demonstrated stable and non-toxic properties [28,29,30,31,33,34,35,36,37,38,42], and fluorinated gas mixtures have also long been used as non-toxic and stable imaging agents [43,44,45,46]. Non-invasive imaging methods such as MRI could obviate the need for biopsies and serve as a biomarker of radiation therapy efficacy or tumor recurrence, thereby improving patient management and outcomes [47]. In this study, a fluorine nano-emulsion was intravenously delivered to mice bearing breast or colorectal tumors to label and, subsequently, track macrophage dynamics in response to radiation therapy [28,47,48]. The cell lines selected for this study were based on their known radio-responsive properties [42,49,50,51,52].

## 2. Materials and Methods

### 2.1. Synthesis of Perfluorocarbon (PFC) Nanoemulsion

The PFC nanoemulsion was prepared in-house. Egg yolk phospholipid (EYP, 0.157 g, Thermo, Waltham, MA, USA.) was added to 4.5 mL deionized water, vortexed briefly, and bath-sonicated. Perfluoro-[15]-Crown-5 Ether (PFCE, 2.205 g, Exfluor, Round Rock, TX, USA.) was added to the mixture and probe-sonicated at 20% power for 2 min. Deionized water was added to bring the total volume to 6.0 mL. The crude emulsion was passed 5 times through a microfluidizer at 20,000 psi with the reaction chamber cooled on ice. The nanoemulsion was sterile-filtered (0.2 µm, Pall, Port Washington, NY, USA.) into autoclaved glass vials, capped, and stored at 4 °C until use. Following formulation and filtration, the nanoemulsion size and concentration was characterized by dynamic light scattering (DLS) and fluorine ^19^F NMR, respectively. 

DLS Malvern Zetasizer Nano ZS90 instrumentation was used to determine the particle size and polydispersity index (PdI) for the samples. Nanoemulsions were diluted to 0.5% *v*/*v* in water and transferred to disposable cuvettes. Measurements were performed in triplicate for each nanoemulsion.

NMR experiments were acquired on a 400 MHz Bruker spectrometer. The ^19^F NMR spectra were acquired using: 17 ms pulse, 32,000 points acquired for free induction decay (FID), 100 ppm spectral width, 64 averages, and a recycle delay of 15 s. For emulsion concentration determination, 50 µL PFC nanoemulsion was added to 750 µL 0.1% (*w*/*v*) Sodium Trifluoroacetate (TFA, Sigma Aldrich, St. Louis, MO, USA.) in D_2_O. PFC concentration was determined by using the relative integrals of the TFA signal and the PFC signal, as described previously [53].

### 2.2. Cell Lines

Murine colon cancer cell line MC38 and murine breast cancer cell line 4T1 were donated from Dr. Sheng Tong at the University of Kentucky in the Department of Biomedical Engineering. Murine macrophage cell line J774 for cytotoxicity assays was donated by Dr. Liangfang Zhang at the University of California San Diego. All cell culture reagents were purchased from Gibco (Waltham, MA, USA). MC38 and J774 were cultured in T75 tissue culture flasks with Dulbecco’s Modified Eagle Medium (DMEM), supplemented with 10% fetal bovine serum (FBS), and 1% penicillin/streptomycin (PS). The 4T1 cell line was cultured in T75 tissue culture flasks with Roswell Park Memorial Institute (RPMI) 1640 medium, supplemented with 10% FBS, and 1% PS. Cell passage took place every two to three days. Cells were allowed to grow in a 37 °C incubator with 5% CO_2_.

### 2.3. Cytotoxicity and Cell Apoptosis Assays

Potential cytotoxicity of the PFC nanoemulsion was analyzed via Cell counting Kit-8 (CCK-8) assay, similar to MTT, and via Annexin-V and Zombie Violet flow analysis [54]. Briefly, for the CCK-8 assay, MC38, 4T1, or J774 cells were plated at a density of 5000 cells per well in a 96-well culture plate. Once attached, dilutions of PFC emulsions of varying concentrations were added to each well, in triplicate (0 μM, 10 μM, and 100 μM final concentration) and the plate was incubated for 24 h at 37 °C and 5% CO_2_. Three wells of RPMI and DMEM media were also prepared for background phenol red measurement. The next day, 10 μL of CCK-8 solution (Cat# CK04, Dojindo, Gaithersburg, MD, USA) was added to each well and the plate was placed back in the incubator for 2 h. Absorbance of each sample was acquired at 450 nm on an Infinite 200 Tecan plate reader. The mean viability of samples compared to control (0 μM) were reported as percentage with standard deviation.

For apoptosis assays using flow cytometry, samples of 5 × 10^5^ cells per cell line were plated in 60 mm^2^ petri dishes in 5 mL final volume. Three dishes of each cell line received 1 mM final concentration of PFC emulsion and three remained untreated. The next day, all wells were harvested into flow cytometry tubes and rinsed in PBS prior to Zombie Violet (Cat# 423113/423114, Biolegend, San Diego, CA, USA) staining per manufacturer protocol. Cells were rinsed with Annexin-V buffer (Cat# 422201, Biolegend) and, subsequently, stained with APC-Annexin V antibody (Cat# 640920, Biolegend). Cells were washed and resuspended for flow processing. Data were acquired on a BD LSRFortessa X-20 and data were processed using FlowJo^TM^.

### 2.4. Mouse Model

All animal experiments were approved by University of Kentucky’s (UK) Institutional Committee on Animal Care and Use (IACUC #2019-3341). Six-to-eight-week-old female Balb/c and C57BL/6J mice were purchased from Jackson Laboratories and given three days for acclimation. Ten Balb/c mice received unilateral mammary fat pad injection of 5 × 10^5^ 4T1 cells in 100 µL PBS (D-5). Ten C57BL/6J mice received unilateral mammary fat pad injection of 5 × 10^5^ MC38 cells in 100 µL PBS. Tumors were allowed to grow for 4–5 days prior to radiation therapy start. Two days prior to radiation (D-2, Figure 1), all mice received intravenous injection of fluorine nanoemulsion (200 µL, C = 150 mg/mL).

### 2.5. MRI Acquisition

Mice were anesthetized with 1–2% isoflurane in O_2_ and positioned supine in a Bruker/Siemens 7T MR Scanner with a dual-tuned ^1^H/^19^F volume coil. Animal temperature was regulated, and respiration was monitored during scans. A reference capillary with dilute ^19^F nanoemulsion was positioned in the image field of view. Then, ^1^H anatomical images were acquired using the TurboRARE sequence with TR/TE = 2000/34 ms., RARE factor 8, matrix 256 × 256, FOV 32 × 32 mm^2^, slice thickness 1 mm, 20 slices, and 1 average. The ^19^F images were also acquired using a TurboRARE sequence with parameters TR/TE = 1800/34 ms. RARE factor 8, matrix 32 × 32, FOV 32 × 32 mm^2^, slice thickness 2 mm, 10 slices, and 180 averages. Baseline MRI was acquired at day 0, before irradiation. Longitudinal MRIs occurred on days 4, 7 or 9, and 15. At the experimental endpoint (day 15), all mice were euthanized via CO_2_ inhalation. Tumor samples were harvested and preserved in 10% Formalin.

### 2.6. Radiation Therapy

After baseline MR imaging, five mice in each tumor model were irradiated with a single 8 Gy dose over an approximately four-minute interval at the X-ray Service Center of the University of Kentucky [55,56]. During therapy, mice were sedated with a ketamine solution (10% ketamine, Covetrus, 5% xylazine, Henry Schein, 85% saline) injected intraperitoneally (100 mg/kg ketamine + 10 mg/kg xylazine at the maximum). Mice were laid supine, with tumors being irradiated locally to the mammary fat pad. The day of RT was defined as day 0 of experimentation. Mice received intraperitoneal (10 µL/g) revertidine solution (5% revertidine, Vedco, 95% saline) to quicken their recovery from anesthesia. One C57BL/6J mouse unintentionally did not recover from anesthesia.

### 2.7. Data Analysis

MR images were analyzed in ImageJ software version 1.53t. The ^1^H/^19^F renderings were performed by overlaying ^1^H (grayscale) and ^19^F (fire) slices. The total number of fluorine atoms in tumor regions was calculated directly from the in vivo ^19^F image using the measure function. Image measurements of the external ^19^F reference capillary signal normalized the reference image. Noise as inputs corrected background noise. Regions of interest (ROIs) were segmented around ^19^F signals in the tumor region. A baseline ^19^F signal for individual mice was established 48 h after injection of the nanoemulsion (D0). From there, changes in ^19^F signal for each mouse were compared to these pre-established values. Only then were mice grouped depending on receival of treatment.

Tumor volume was calculated from MR images using an elliptical cone shape formula. Necessary dimensions for volume calculation were determined by using the proton slice with the greatest tumor size and measuring two orthogonal axes. The third axis was determined through summation of the number of slices where tumor was present and multiplied by set distance between slices.

### 2.8. Statistical Analysis

Descriptive summaries of the data were presented overall and by cancer cell line types (4T1 and MC38), by treated versus untreated groups, and at each follow-up time point. These summaries include the mean, median, and standard error, and are reported overall and by groups at each time point. We employed a general linear model (ANOVA) to compare group mean estimates at each time point. Additionally, we utilized a generalized linear model, to account for the repeated measurement of the observations, and incorporated the compound symmetric covariance structure for model fitting. All statistical hypothesis tests were conducted using the standard 5% significance level. To assess differences between the two cell line types concerning normalized fluorine signal and tumor volume, and treated versus untreated groups at each time point, we applied the Kruskal–Wallis test, a non-parametric statistical approach. Additionally, we calculated correlation coefficients, both overall and for individual time points, to elucidate the relationship between normalized fluorine signal and tumor volume, and treated versus untreated groups. SAS software Version 9.4 (TS1M1, SAS Institute, Cary, NC, USA) was used for all analyses.

## 3. Results

### 3.1. F MRI Allows for Longitudinal Tracking of Macrophages in Tumors following RT

The fluorine nanoemulsion used in this project had a mean droplet size of 149.6 ± 0.56 nm, a PdI of 0.013 ± 0.002, and a concentration of 150 mg/mL. The characterization of several of our recent formulations is summarized in Table 1. The emulsion was stable throughout the course of the experiment (size variation < 10% and PdI < 0.2). Cytotoxicity experiments revealed no significant viability impairment at both concentrations tested (Appendix A). Similarly, the PFC emulsion did not incur any increase in dead or apoptotic cell numbers, as suggested by the Zombie Violet and Annexin-V stains, respectively (Appendix A). As previously described, the systemic delivery of fluorine emulsions labels phagocytic cells and, more specifically, tumor-associated macrophages in the TME with the peak uptake within 48 h [26,28]. This corresponded to our ‘day 0′ time point, with baseline ^19^F MRIs being acquired prior to RT treatment (Figure 1). At baseline, all mice exhibited the TAM-associated fluorine signal in colon tumors (Figure 2). The signal appeared to increase in the treated group (Figure 2A), while remaining stable or diminishing in the untreated group (Figure 2B). As a side note, the ^19^F signal was also observed in the spleen, liver, lymph node, and bone marrow to some extent due to resident macrophages in those tissues and standard fluorine clearance mechanisms. This is coherent with numerous other reports using fluorinated agents [26,28,57,58,59].

### 3.2. Colon RT-Treated Tumors Show Strong Macrophage Influx and Growth Arrest

As expected, the colon tumors displayed significant tumor growth reduction following a high-dose radiation therapy as early as 7 days post-RT (* *p* = 0.01, Figure 3A). Tumors indeed appeared to undergo a complete growth arrest over the two-week-long experiment. Conversely, the untreated mice experienced severe tumor growth within that time frame (Day 15 * *p* = 0.003, Figure 3A). Normalized fluorine signal measurements in tumors increased significantly in the RT-treated group as early as day 4 post-treatment (* *p* = 0.01, Figure 3B) and continued to remain significantly higher than the untreated group at days 7 and 15 (* *p* = 0.05 and * *p* = 0.02, respectively). This indicates a significant macrophage influx in the treated tumors, which concurs with previous literature [39,60]. The untreated mice displayed a slight increase in fluorine signal (Figure 3B) over two weeks’ time, likely due to the expected ongoing recruitment of tumor-associated macrophages to a rapidly growing tumor.

### 3.3. Breast RT-Treated Tumors Show Moderate Fluorine Signal Increase and Growth Reduction

In the breast cancer model, the baseline (day 0) fluorine signal was higher for all mice compared to the colon cancer model (Figure 4). In the treated group, the signal intensity appeared to persist or increase with respect to tumor growth (Figure 4A), whereas the signal decreased in the control group (Figure 4B). Conversely to the colon cancer model, all mice bearing 4T1 tumors displayed slower growth but persistent growth, even after therapy. Quantitatively, tumor growth reduction in the RT-treated mice was significant as early as day 4 post-therapy (* *p* = 0.009, Figure 5A) and remained significantly different throughout the experiment. The fluorine signal in the RT-treated mice increased slightly over the two weeks’ course while the untreated mice experienced significant signal loss (Figure 5B). In 4T1 tumors, the macrophage influx seemed slower compared to the MC38 model, as a significant signal change only occurred 15 days post-RT (* *p* = 0.0005, Figure 3B and Figure 5B).

### 3.4. Macrophage Signal Correlates to Tumor Growth

To gauge the power of our imaging method, we ran correlation tests between the fluorine signal detected and tumor burden irrespective of the time point. Spearman’s correlation coefficients were significant for all treatment groups (*p* < 0.03, Table 2, Figure 6). Since Spearman’s method accounts for a normal distribution of the data and because of the limited sample size, we also evaluated the correlation through Kendall Tau’s method (Table 2). In that case, three out of four treatment groups showed a significant correlation, and the 4T1 treated group reached a *p*-value of 0.077. This may, in part, be due to the delayed influx of macrophages to the tumor post-therapy.

## 4. Discussion

This study effectively reinforced MRI’s ability to track macrophages in the tumor microenvironment. A significant change in ^19^F signal was found between radiotherapy-treated and untreated groups for both the breast and colon cancer models two weeks after irradiation. Treated groups in both cohorts also displayed significant tumor regression. The fluorine signal change correlated to tumor volumes, indicating that the MR imaging of macrophage recruitment in tumors may be a valid biomarker of RT efficacy.

The observed tumor regression from RT for both breast and colon cancers concurs with existing literature [42,49,51,52,61,62]. Other studies have also demonstrated the power of MRI to track macrophages, thereby achieving a better understanding of the TME [28,47,48,60]. Specifically, Croci et al. showed that ^19^F MRI signal increases 7 days post single 10 Gy RT, consistent with our findings [39]. However, this study is the first to correlate the macrophage signal change to tumor volume and a first step towards an accurate non-invasive assessment of RT efficacy.

Fluorinated nanoemulsions have been used in research and clinical applications for over three decades and exhibit excellent biological and chemical safety profiles [63]. Their oxygen-dissolving properties made them attractive as a primary compound in blood substitutes and tumor oxygenation measurements in the 1990s [64,65,66]. Nanoemulsions were then developed in the early 2000s to label and track specific cells such as stem cells, immune cells, and cancer cells in multiple disease models [25,29,30,34,35,36,37,38,50,67], and was especially useful in cancer monitoring [26,27,28,31,39]. None of these studies reported cell toxicity or liver toxicity when injected systemically. These have since been investigated in clinical trials [40,41,68] with promising results.

Oncologists, radiation oncologists, radiologists, and other medical experts specializing in cancer care currently lack an effective, noninvasive technique for the longitudinal quantification of TAMs. At present, they depend on tissue biopsies and pathology examinations to evaluate the effectiveness of therapy. Although accurate to some degree, technical feasibility, tumor heterogeneity, and cancer evolution limit this approach [69]. Treatment efficacy can also be assessed through clinical imaging techniques but rely on the radiologists’ expertise and may be obscured by pseudo-progression mechanisms [70]. The use of biocompatible fluorinated nanoemulsions in MRI not only provides anatomical location detail, but also quantitative data of TAMs. Indeed, quantifiable metrics are extremely important for reducing uncertainty in diagnosis and analyzing treatment efficacy. Furthermore, the non-invasive characteristic of this approach has the potential to reduce the necessity for tumor biopsies and allows for the assessment of treatment effectiveness at different time intervals.

Tumors possess apparent mechanistic differences depending on their origin [71] and have, therefore, varied responses to RT [5]. Therefore, it was imperative to analyze more than one tumor model to acquire a better understanding of differences in radiation-induced TAM involvement. The relationship between TAM recruitment and tumor growth varied between the two groups. Initially, in the colon cancer model, TAM quantities started low relative to tumors in the breast cancer model and increased slowly once treated. In contrast, 4T1 TAM quantities started higher but degraded over the trial if they were not exposed to radiation. The relationship between growth and TAM involvement is evidently tumor-specific. This finding can perhaps speak to other properties of tumors such as aggressiveness and its association with TAM burden [72].

This method of tracking macrophages remains limited by the lack of knowledge on the specific and dynamic macrophage phenotype. As shown in other studies, complex interactions occurring in the TME govern anti-tumorigenic and pro-tumorigenic inflammation processes [2,3,39]. Anti-inflammatory (M2) tumor-associated macrophage (TAM) recruitment in tumors is an established biomarker of tumor aggressiveness and resistance to therapy [28,60,73,74,75,76,77], and early recurrence following radiotherapy in both breast and colorectal cancers [78,79,80]. The extent and timing of the differentiation from tumor inhibiting to tumor promoting is widely unknown [81,82]. Specific M1 or M2 targeted probes may one day help decipher phenotype dynamics. Additionally, although the utilization of two distinct murine cancer models and mouse strains served as an excellent foundation for showcasing the capabilities of this imaging technique, it is worth noting that these cell lines were known for that radiosensitivity. Experiments involving cell lines of varying radio-resistivity will provide further insights. Additionally, an investigation into how varying doses of radiation and possible fractionation affect macrophage recruitment, as well as extending the period of the imaging time, will provide a deeper understanding of inflammation’s role in tumor recurrence [72].

## 5. Conclusions

In conclusion, this study demonstrated the correlation between increased macrophage infiltration in RT-treated tumors, as assessed via non-invasive ^19^F MRI, and tumor growth reduction. Our findings support the potential therapeutic benefit of incorporating MRI of macrophage dynamics in the RT workflow. Non-invasive imaging tools are becoming increasingly necessary in order to provide a safe and accurate tumor prognosis. Moreover, quantitative data that represent interactions occurring in the TME provide medical professionals with a deeper understanding of treatment efficacy. By integrating state-of-the-art algorithms into clinical settings, the processing of these data will not only become more precise but also significantly faster. Finding ways to harness this powerful information will be the next challenge, but, ultimately, can lead to dramatically better patient outcomes.

## Figures and Tables

**Figure 1 cancers-15-05874-f001:**
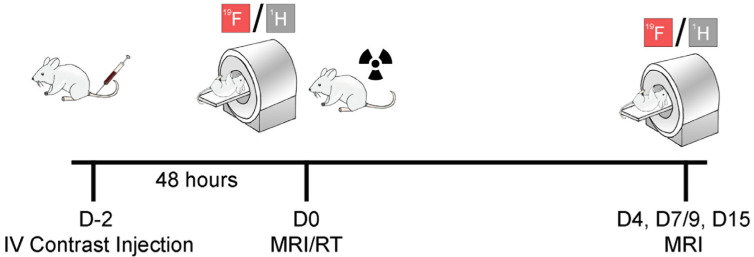
Experimental design to monitor macrophage dynamics after RT via ^19^F MRI: Breast- or colon-tumor-bearing mice received fluorine nanoemulsion injection IV two days prior to baseline MR imaging (D0). Mice then underwent a single 8 Gy RT dose (D0) and longitudinal ^1^H/^19^F MRIs were acquired on days 4, 7 or 9, and 15 to quantify ^19^F macrophage signal and tumor volume change.

**Figure 2 cancers-15-05874-f002:**
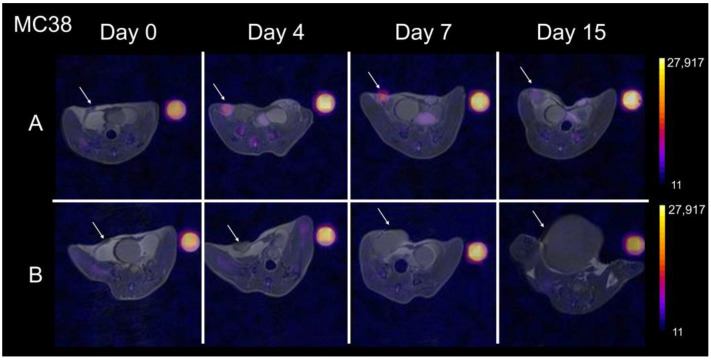
Longitudinal in vivo imaging of ^19^F-labeled macrophages in colon tumors: Representative ^1^H/^19^F MRI overlays of RT-treated (**A**) and untreated (**B**) control mouse bearing MC38 tumors at day 0, 4, 7, and 15 after RT. The treated group (**A**) showed modest tumor growth (arrow) and increased ^19^F signal over time. Contrarily, the untreated group (**B**) shows rapid tumor growth and unchanged fluorine signal over 15 days. The color scale is in arbitrary units.

**Figure 3 cancers-15-05874-f003:**
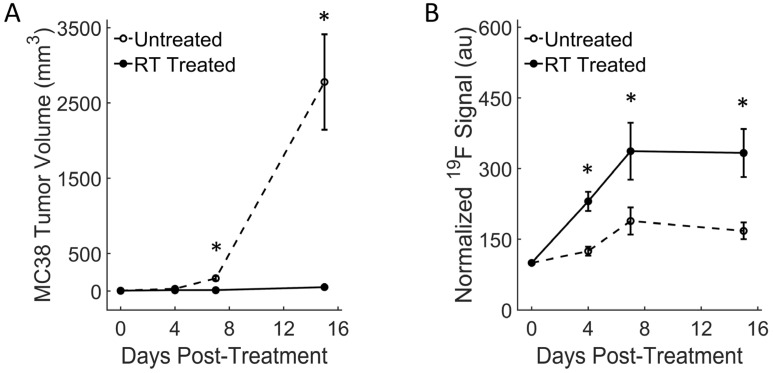
Quantitative analysis of macrophage dynamics in colon tumors: (**A**) Longitudinal tumor volume measurements show significant tumor growth reduction in the RT-treated MC38 tumors 7 days post-treatment (* *p* = 0.01) compared to untreated. In comparison, tumor growth is very rapid in the untreated group. By day 15, the gap between groups widens further (* *p* = 0.003). (**B**) Normalized fluorine signal in tumors increases significantly in the RT-treated group as early as day 4 post-treatment (* *p* = 0.01) and continues to remain significantly higher than the untreated group (* *p* = 0.05 and * *p* = 0.02 at days 7 and 15, respectively). This indicates significant macrophage influx in the treated tumors. Data are presented as mean ± standard error.

**Figure 4 cancers-15-05874-f004:**
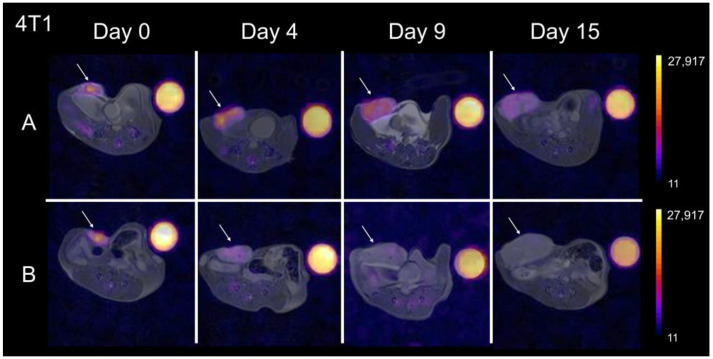
Longitudinal in vivo imaging of ^19^F-labeled macrophages in breast tumors: Representative ^1^H/^19^F MRI overlays of RT- treated (**A**) and untreated (**B**) control mouse bearing 4T1 tumors at day 0, 4, 9, and 15 after RT. The treated group (**A**) showed modest tumor growth (arrow) and increased ^19^F signal over time. Contrarily, the untreated group (**B**) shows faster tumor growth and fluorine signal decrease over 15 days. The color scale is in arbitrary units.

**Figure 5 cancers-15-05874-f005:**
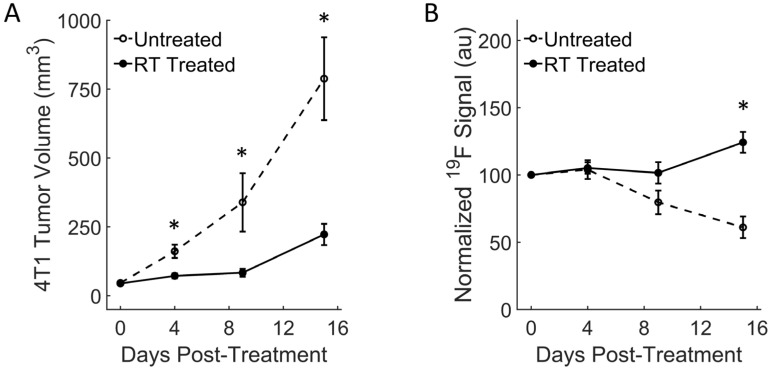
In vivo quantification of macrophage dynamics in breast tumors. (**A**) Longitudinal tumor volume measurements show significant tumor growth reduction in the RT-treated 4T1 tumors 4 days post-treatment (* *p* = 0.009). By day 15, tumor growth reduction continues (* *p* = 0.009). (**B**) In 4T1 tumors, the fluorine signal increases more moderately and is significantly different from untreated tumors by day 15 (* *p* = 0.0005). In the untreated tumors, the fluorine signal steadily decreases over the two-week period. Data are presented as mean ± standard error.

**Figure 6 cancers-15-05874-f006:**
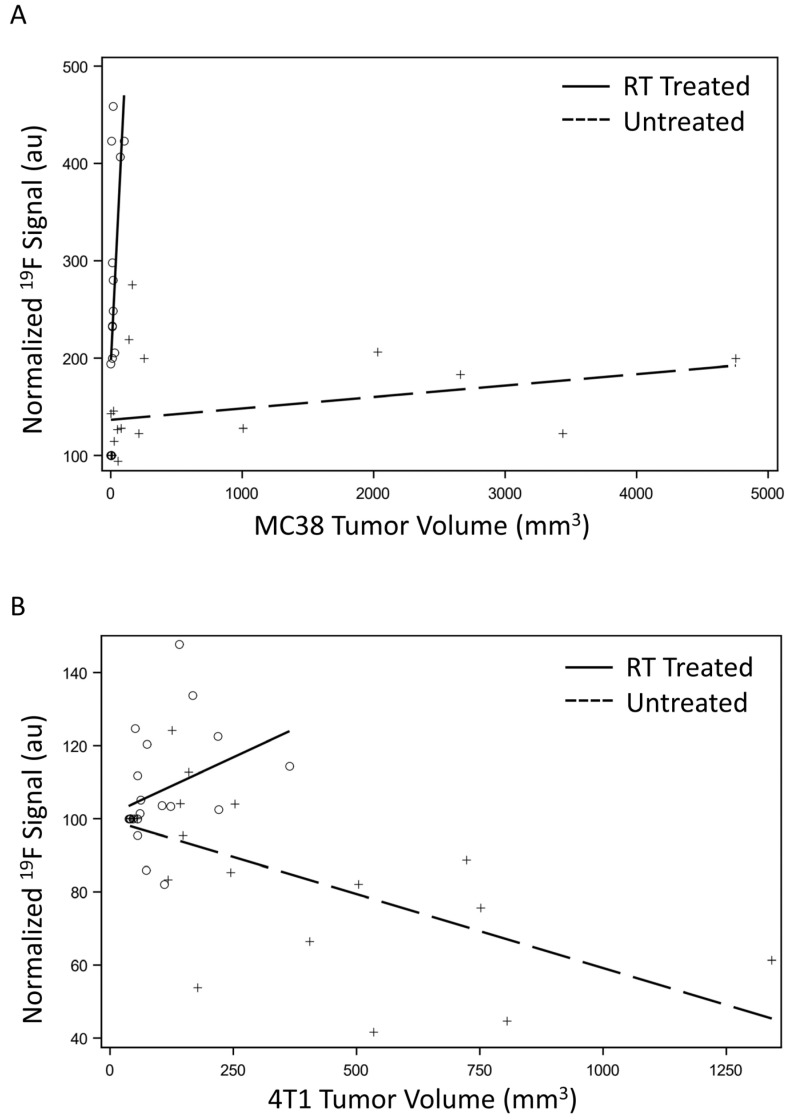
Correlation of macrophage signal to tumor growth: Kendall Tau correlation tests show significant correlation between the normalized fluorine signal and tumor volume with respect to treatment in MC 38 (**A**) and 4T1 tumors (**B**), independent of time.

**Table 1 cancers-15-05874-t001:** Dynamic light scattering data of the perfluorocarbon emulsions.

Batch	Size (nm)	PDI
1	179.4	0.158
2	175.7	0.144
3	139.0	0.010
4	179.8	0.142
5	173.2	0.150
6	149.6	0.013
Standard Deviation	±17.4	±0.071

**Table 2 cancers-15-05874-t002:** Correlation coefficients and associated p-values comparing normalized fluorine signal and tumor volume in each group.

Correlation Coefficient (*p*-Value)	MC38		4T1	
	Treated	Untreated	Treated	Untreated
Spearman	0.710 (0.002)	0.575 (0.008)	0.474 (0.035)	−0.644 (0.002)
Kendall Tau	0.528 (0.006)	0.405 (0.015)	0.292 (0.077)	−0.454 (0.006)

## Data Availability

The data that support the findings of this study are available from the corresponding author upon request.

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
