# Peer review of "Magnetic Resonance Imaging of Macrophage Response to Radiation Therapy"

_cancers, 2023, doi:10.3390/cancers15245874_

Round 1
Reviewer 1 Report
Comments and Suggestions for Authors
Comments to the authors
The authors (Yang et al) have developed Magnetic resonance imaging of macrophage response to radiation therapy. However, there are some points which need to be taken care of. Following are some of the comments that the authors might find useful for future submission. The manuscript should be revised before publication.
Comment:
1. The author has prepared the perfluorocarbon (PFC) nano-emulsion for the application at site. The author should clearly mention the benefit of choosing the nano-emulsion as their final formulation and how it could be applied clinically. The author has also not mentioned anything about sterilization of the of formulation which can be a critically important.
- The author is encouraged to provide a comprehensive exposition of the Dynamic Light Scattering (DLS) data, including the standard error values, for each of the formulations to enhance the rigor and transparency of the study.
3. The author needs to explain stability of nano-emulsion.
- The choice of breast and colon models as subjects for this study necessitates clarification. The rationale for selecting these specific models should be explicitly stated to enhance the scientific integrity of the research.
- It is essential to include studies evaluating the potential toxicity of the formulation on healthy cells to provide a comprehensive safety assessment, which is currently missing.
- The author should elaborate on the methods and techniques used to quantitatively measure and verify the quantity of the formulated product to ensure transparency and reproducibility in the research.
Comments on the Quality of English Language
It is good in writing
Reviewer 2 Report
Comments and Suggestions for Authors
In the current study, the authors evaluated the tracking of macrophages in tumors following radiation therapy using 19F MRI technology. This is a very interesting study. However I have the following questions to be answered or corrected.
(1) We do not know the in vivo or in vitro stability of PFC nano-emulsion. Is there a QC test for the PFC nano-emulsion, please provide the exact parameter of the PFC nano-emulsion, for example concentration.
(2) It is said that tumor associated macrophage (TAM) recruitment in tumors is an biomarker of tumor aggressiveness and resistance to therapy and early recurrence following radiotherapy in both breast and colorectal cancers. The study results showed that in radiation group there is an increased 19F signal over time which means aggressiveness, but the tumor size reduced. So tt seems that the result and the explanation are somewhat contradictory, please explain it.
(3) For figure 2, is there an intensity scale for picture? Can you provide one whole body picture to show the distribution?
(4) It would be better if there is the immunostaining of tumor, which can explain if the higher intensity signal related with the TAM.
Round 2
Reviewer 1 Report
Comments and Suggestions for Authors
- It is essential to include studies evaluating the potential toxicity of the formulation on healthy cells to provide a comprehensive safety assessment, which is currently missing.
- Author can refer this paper (https://pubs.acs.org/doi/full/10.1021/acsami.9b16648)
I would suggest that the manuscript requires only minor enhancements in terms of English language quality.
Reviewer 2 Report
Comments and Suggestions for Authors
The authors answered all the questions. And the revised manuscript has been sufficiently improved for publication.
Author Response
We'd like to extend our thanks to the reviewer for their insightful revisions.